# Optimal Sampling and Clustering in the Stochastic Block Model

**Se-Young Yun**
KAIST
Daejeon, South Korea
yunseyoung@kaist.ac.kr

**Alexandre Proutière**
KTH
Stockholm, Sweden
alepro@kth.se

## Abstract

This paper investigates the design of joint adaptive sampling and clustering algorithms in networks whose structure follows the celebrated Stochastic Block Model (SBM). To extract hidden clusters, the interaction between edges (pairs of nodes) may be sampled sequentially, in an adaptive manner. After gathering samples, the learner returns cluster estimates. We derive information-theoretical upper bounds on the cluster recovery rate. These bounds actually reveal the optimal sequential edge sampling strategy, and interestingly, the latter does not depend on the sampling budget, but on the parameters of the SBM only. We devise a joint sampling and clustering algorithm matching the recovery rate upper bounds. The algorithm initially uses a fraction of the sampling budget to estimate the SBM parameters, and to learn the optimal sampling strategy. This strategy then guides the remaining sampling process, which confers the optimality of the algorithm. We show both analytically and numerically that adaptive edge sampling yields important improvements over random sampling (traditionally used in the SBM analysis). For example, we prove that adaptive sampling significantly enlarges the region of the SBM parameters where asymptotically exact cluster recovery is feasible.

## 1 Introduction

Extracting clusters in networks is a central task in many fields including biology, computer science, and social science. The Stochastic Block Model (SBM) [9] and its extensions provide a natural statistical benchmark to assess the performance of network clustering algorithms. The SBM defines a random graph with $n$ nodes and consisting of $K$ non-overlapping clusters, $\mathcal{V}_1, \ldots, \mathcal{V}_K$, of respective sizes $\alpha_1 n, \ldots, \alpha_K n$ with $\alpha_k > 0$ for all $k$. An edge between two nodes from respective clusters $\mathcal{V}_i$ and $\mathcal{V}_j$ indicates whether these nodes interact and appears in the graph with probability $p_{ij}$, independently of other edges. The SBM is hence parametrized by $\boldsymbol{p} = [p_{ij}]_{1 \leq i,j \leq K}$ and $\boldsymbol{\alpha} = (\alpha_1, \ldots, \alpha_K)$. We assume that the relative cluster sizes $\boldsymbol{\alpha}$ do not depend on the network size $n$, whereas on the contrary, $\boldsymbol{p}$ may vary with $n$. Most existing work on the SBM and its extensions investigate the problem of recovering the clusters from an observed realization of the random graph.

In contrast, in this paper, we are interested in active learning scenarios where the interaction between pairs of nodes may be sampled sequentially, which allows a given node pair to be sampled several times. In these scenarios, the algorithm samples edges in an adaptive manner: In a given round, the edge selected to be sampled may depend on the information gathered previously, and should the algorithm select the edge $(v, w) \in \mathcal{V}_i \times \mathcal{V}_j$, it observes a Bernoulli r.v. with mean $p_{ij}$, independent of the previous observations. The algorithm has an observation budget of $T$ samples (typically depending on the network size), and after collecting these samples, it should return estimates of the clusters. The objective is to devise a joint sampling and clustering algorithm such that the estimated clusters are as accurate as possible. Specifically, we aim at characterizing the minimal cluster recovery

error rate for a given observation budget $T$. Adaptive sampling can be critical in clustering tasks where collecting edge samples is expensive (e.g., in biology, one has to run tedious experiments to assess whether two proteins share similarities). For such tasks, it is important to discover clusters with a minimum number of samples (these in turn need to be selected in an adaptive manner).

Even for the simple binary symmetric SBM (i.e., two clusters of equal sizes) with non-adaptive sampling, obtaining an explicit expression for the minimal number of mis-classified nodes remains illusory, especially when the graph is sparse, i.e., when the $p_{ij}$'s scale as $1/n$ (see e.g. [6, 14]). Hence, in this paper, we restrict our attention to models where only a vanishing fraction of nodes is allowed to be mis-classified. More precisely, for any $s = o(n)$, we aim at identifying a necessary and sufficient condition on $n$, $T$, $p$ and $\alpha$ such that there exists a joint adaptive sampling and clustering algorithm mis-classifying less than $s$ nodes with high probability. This objective is more ambitious than just deriving conditions for *weak consistency* (also referred to as asymptotically accurate detection) [15, 12, 1], that is to say, conditions under which the proportion of mis-classified nodes vanishes as $n$ grows large. Indeed, we are interested in the minimal recovery error rate. Further observe that deriving conditions for asymptotically exact recovery is part of our objective (these conditions are obtained selecting $s = 1$).

**Main results.** We establish that under mild assumptions, for any $s = o(n)$, a necessary and sufficient condition for the existence of a joint adaptive sampling and clustering algorithm mis-classifying less than $s$ nodes w.h.p.[1] is

$$\liminf_{n \to \infty} \frac{2TD(\boldsymbol{p}, \boldsymbol{\alpha})}{n \log(n/s)} \geq 1, \tag{1}$$

where the *divergence* $D(\boldsymbol{p}, \boldsymbol{\alpha})$ is defined as: $D(\boldsymbol{p}, \boldsymbol{\alpha}) = \max_{\boldsymbol{x} \in \mathcal{X}(\boldsymbol{\alpha})} \Delta(\boldsymbol{x}, \boldsymbol{p})$,

with $\Delta(\boldsymbol{x}, \boldsymbol{p}) = \min_{i,j:i \neq j} \sum_{k=1}^{K} x_{ik} KL(p_{ik}, p_{jk})$ and

$$\mathcal{X}(\boldsymbol{\alpha}) = \left\{ \boldsymbol{x} = [x_{ij}] : \alpha_i x_{ij} = \alpha_j x_{ji}, \ \sum_{i=1}^{K} \alpha_i \sum_{j=1}^{K} x_{ij} = 1, \ \text{and} \ x_{ij} \geq 0, \ \forall i, j \right\},$$

and where $KL(a, b)$ denotes the KL divergence between two Bernoulli distributions with respective means $a$ and $b$. A consequence of this result is that when $T = \omega(n)$, the best possible joint sampling and clustering mis-classifies $n \exp(-\frac{2T}{n} D(\boldsymbol{p}, \boldsymbol{\alpha})(1 + o(1))$ nodes.

**Gains through adaptive sampling: exact recovery conditions and numerical experiments.** To illustrate the gain obtained by adaptive sampling, we can compare the conditions for asymptotically exact recovery with or without adaptive sampling. Consider the following binary symmetric SBM: $K = 2$, $\boldsymbol{\alpha} = (1/2, 1/2)$, $p_{11} = p_{22} = \frac{af(n)}{n}$, and $p_{12} = p_{21} = \frac{bf(n)}{n}$. For the classical cluster recovery problem in SBMs without adaptive sampling, one observes a realization of the random graph and hence one has $T = n(n-1)/2$ observations (one per pair of nodes) to estimate clusters. For the above binary symmetric SBM, asymptotically exact recovery [3] is possible if and only if either $f(n) = \omega(\log(n))$ or $f(n) = \log(n)$ and

$$\max\{\sqrt{a} - \sqrt{b}, \sqrt{b} - \sqrt{a}\} > \sqrt{2} \quad \text{(Non-adaptive sampling)}$$

Now with adaptive sampling and the same observation budget $T = n(n-1)/2$, our results show that asymptotically exact recovery is feasible if and only if either $f(n) = \omega(\log(n))$ or $f(n) = \log(n)$ and

$$\max\{a \log(\frac{a}{b}) + b - a, b \log(\frac{b}{a}) + a - b\} > \frac{1}{2} \quad \text{(Adaptive sampling)}$$

Figure 1 (left) presents the regions (described through $a$ and $b$) where asymptotically exact recovery is feasible with or without adaptive sampling. Observe that adaptive sampling significantly enlarges the region where exact recovery is possible.

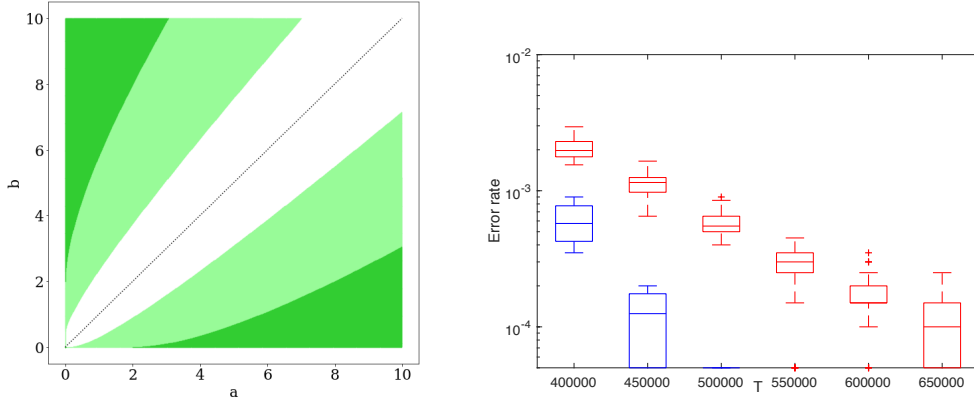

Figure 1: (Left) Regions where asymptotically exact recovery is possible for the binary symmetric SBM. The intra- and inter-cluster edge probabilities are $a \log(n)/n$ and $b \log(n)/n$. Dark green: region with non-adaptive sampling, dark green+light green: region with adaptive sampling. (Right) Recovery error rate of ASP (with adaptive sampling) in blue and of the optimal clustering algorithm with random sampling [16] in red for the binary symmetric SBM with $n = 20000$ nodes and $p_{11} = 0.5$ and $p_{12} = 0.1$ – Figure done using matlab boxplot function with outliers (the red crosses).

To further illustrate the gain achieved with adaptive sampling, we compare Figure 1 (right) the cluster recovery rate using ASP (Adaptive Sampling Partition), the proposed optimal joint sampling and clustering algorithm and the optimal clustering algorithm with random sampling presented in [16]. This experiment concerns the above binary symmetric SBM with 20000 nodes and parameters $p_{11} = p_{22} = 0.5$, and $p_{12} = p_{21} = 0.1$. As soon as the sampling budget exceeds 450000, ASP exactly recovers the clusters, whereas a much larger budget is needed to achieve exact recovery if edges are sampled randomly.

**Deriving fundamental limits.** Existing information-theoretic limits derived in the classical SBM without adaptive sampling concern the *expected* number of mis-classified nodes [16, 7]. Here, we are interested in establishing lower bounds on the number of mis-classified nodes that hold with high probability, which is more challenging than establishing similar bounds in expectation. To derive the necessary condition (1), we leverage and combine change-of-measure arguments as those used in online stochastic optimization [10], as well as tools from hypothesis testing. In particular, we need to consider and enumerate a large number of hypotheses pertaining to the way nodes are allocated to the various clusters (these hypotheses concern allocations that differ from more than $s$ nodes). Such an enumeration is not required to derive lower bounds on the expected number of mis-classified nodes [16]. There, simple symmetry arguments can be exploited instead.

**An explicit optimal sampling strategy.** As in some other sequential decision making problems (e.g. bandit problems), the fundamental limits do not only provide the performance of the best possible algorithm, but also provide insights into the design of such an algorithm. To devise a joint sampling and clustering strategy whose performance matches our fundamental limits, we first exploit the following interpretation of the divergence $D(\boldsymbol{p}, \boldsymbol{\alpha}) = \max_{\boldsymbol{x} \in \mathcal{X}(\boldsymbol{\alpha})} \Delta(\boldsymbol{x}, \boldsymbol{p})$ involved in our necessary condition. There, the vector $\boldsymbol{x}$ encodes the average number of samples of edges between the various clusters. More precisely, $2x_{ij}T/n$ is the average number of samples of edges between a given node in cluster $\mathcal{V}_i$ and nodes in cluster $\mathcal{V}_j$. With this interpretation, an optimal sampling strategy consists in allocating the observation budget $T$ according to $\boldsymbol{x}^*(\boldsymbol{p}, \boldsymbol{\alpha}) = \arg\max_{\boldsymbol{x} \in \mathcal{X}(\boldsymbol{\alpha})} \Delta(\boldsymbol{x}, \boldsymbol{p})$. Note that interestingly, $\boldsymbol{x}^*(\boldsymbol{p}, \boldsymbol{\alpha})$ does not depend on the total observation budget. However, the optimal budget allocation depends on the initially unknown SBM parameters $(\boldsymbol{p}, \boldsymbol{\alpha})$. To devise an optimal joint sampling and clustering algorithm, we start, using a fraction of the observation budget, by estimating $\boldsymbol{x}^*(\boldsymbol{p}, \boldsymbol{\alpha})$. More precisely, the proposed algorithm consists in three main steps: (i) first, we use a small fraction of the observation budget and spectral methods to obtain initial cluster estimates; (ii) the latter are then used to derive precise estimators of the SBM parameters, which in turn yield an estimate $\hat{\boldsymbol{x}}^*$ of $\boldsymbol{x}^*(\boldsymbol{p}, \boldsymbol{\alpha})$; (iii) finally, $\hat{\boldsymbol{x}}^*$ and our initial cluster estimates dictate the way to sample edges with the remaining budget, and based on these observations, the cluster estimates are improved.

## 2 Related Work

Clustering in the SBM and its extensions have received a lot of attention recently. Almost all studies concern the problem of recovering the clusters from a realization of the random graph generated under the SBM (one sample for each edge is observed). Nevertheless, it is interesting to summarize the results obtained in this simple non-adaptive setting. Results may be categorized depending the targeted level of performance.

The lowest level of performance is often referred to as *detectability* and requires that the extracted clusters should be only positively correlated with the true clusters. In fact, the question of detectability is mostly relevant in the case of the sparse SBM, where the intra- and inter-cluster edge probabilities $p$ and $q$ scale as $1/n$, say $p = \frac{a}{n}$, $q = \frac{b}{n}$. In the case of the binary symmetric SBM, it has been established that a necessary and sufficient condition for detectability was $a - b > \sqrt{2(a+b)}$ [5, 13, 11]. Refer to [1] for more recent results.

Asymptotically accurate recovery or weak consistency refers to scenarios where the proportion of mis-classified nodes vanishes as $n$ grows large. Necessary and sufficient conditions for such recovery have been derived in [15, 12, 4]. Recently, the authors of [17, 7, 16] manage to quantify the optimal recovery rate when asymptotically accurate recovery is possible. Unfortunately, in these papers, the authors establish a lower bound for the *expected* number of mis-classified nodes and provide an algorithm with guarantees valid with high probability. In this paper, we fix this gap, and develop new techniques to derive lower bounds valid with high probability.

The highest level of performance, asymptotically exact recovery, means that there is no mis-classified node asymptotically with high probability. Necessary and sufficient for exact recovery are provided in [2, 12, 8, 16].

In this paper, we cover both asymptotically accurate detection, and exact recovery. But unlike the aforementioned papers, we investigate the design of joint adaptive sampling and clustering algorithms. As far as we are aware, the only relevant reference for adaptive sampling in the SBM is [15], and only provides a condition for asymptotically accurate detection in homogeneous SBMs where the intra- and inter-cluster edge probabilities do not depend on the clusters (i.e., $p_{ii} = p$ and $p_{ij} = q$ for all $i \neq j$). We manage to derive matching lower and upper bounds valid with high probability on the recovery rate for general SBMs. Our algorithm is very different than that developed in [15] since it is based on the explicit optimal sampling strategy revealed by our lower bounds on the cluster recovery rate (lower bounds that are lacking in [15]).

## 3 Fundamental Limits

This section is devoted to state and prove a necessary condition for the existence of a joint sampling and clustering algorithm mis-classifying less than $s = o(n)$ nodes with high probability. The derivation of the necessary condition combines hypothesis testing techniques and change-of-measure arguments where we pretend that the observations are generated by models obtained by slightly modifying the true SBM model. More precisely, modified models are built by moving nodes from one cluster to another. Since the clusters have different sizes, the number of nodes moved from cluster $\mathcal{V}_i$ to cluster $\mathcal{V}_j$ should depend on $i$ and $j$. As a consequence, the different resulting models should have different distribution vectors $\boldsymbol{\alpha}$. To deal with this asymmetry, we hence introduce the class of $(s, \beta)$-locally stable algorithms defined as follows.

**Definition 1** (($s, \beta$)-*locally stable algorithms*)**.** *A joint sampling and clustering algorithm $\pi$ is $(s, \beta)$-locally stable at $(\boldsymbol{p}, \boldsymbol{\alpha})$, if there exists a sequence $\eta_n \geq 0$ with $\lim_{n\to\infty} \eta_n = 0$ such that for all partition vectors $\tilde{\boldsymbol{\alpha}}$ such that $\|\tilde{\boldsymbol{\alpha}} - \boldsymbol{\alpha}\|_2 \leq \beta$, $\pi$ mis-classifies at most $s$ nodes with probability greater than $1 - \eta_n$ for any $n$. (Note that the definition makes sense even if $\boldsymbol{p}$ depends on $n$.)*

We derive our necessary condition for $(s, \beta)$-locally stable algorithms. Considering $(s, \beta)$-locally stable algorithms is not restrictive, as *good* algorithms should adapt to the SBM and in particular, to various possible proportions of nodes in the different clusters. Furthermore, the theorem below is valid for all $\beta \geq \frac{s}{n} \log\left(\frac{n}{s}\right)$, and hence $\beta$ can be made as small as we want when $n$ grows large.

**Theorem 1.** *Let $s = o(n)$. Assume that there exists a $(s, \beta)$-locally stable clustering algorithm at $(\boldsymbol{p}, \boldsymbol{\alpha})$ for $\beta \geq \frac{s}{n} \log\left(\frac{n}{s}\right)$. Then we have:* $\liminf_{n\to\infty} \frac{2TD(\boldsymbol{p},\boldsymbol{\alpha})}{n\log(n/s)} \geq 1.$

To establish Theorem 1, we consider a $(s, \beta)$-locally stable algorithm, and assume that the corresponding budget allocation is defined by $\boldsymbol{x}$ (representing the expected number of samples gathered from a node to all clusters). We then exhibit a large number $M$ of hypotheses, each defined by an allocation of nodes to clusters. These hypotheses correspond to allocations differing from each other by more than $s$ nodes. We enumerate the hypotheses and quantify $M$ as a function of $n$ and the SBM parameters. Using the fact that the algorithm is $(s, \beta)$-locally stable, we can find a *worst* hypothesis (not corresponding to the true allocation of nodes) occurring with probability less than $\eta_n/M$. Next, we apply a change-of-measure argument. Specifically, we pretend that the observations are generated by a (perturbed) allocation built from that corresponding the worst hypothesis. We study the log-likelihood ratio of the observations under the true and the perturbed allocations. Combining the analysis with the fact that the worst hypothesis occurs with probability less than $\eta_n/M$, we conclude that the number of nodes from $\mathcal{V}_j$ actually classified in $\mathcal{V}_i$ must roughly exceed $\alpha_j n \exp(-\frac{2T}{n} \sum_{k=1}^{K} x_{ik} KL(p_{ik}, p_{jk}))$. This implies that at least $n \exp(-\frac{2T}{n} \Delta(\boldsymbol{x}, \boldsymbol{p}))$ nodes are mis-classified. Now optimizing this lower bound over $\boldsymbol{x}$, we deduce that at least $n \exp(-\frac{2T}{n} D(\boldsymbol{p}, \boldsymbol{\alpha}))$ nodes are mis-classified. The complete proof is presented in Appendix. There, we also provide the proof for the binary symmetric SBM (this proof helps the understanding of that for general SBMs).

# 4 The Adaptive Spectral Partition Algorithm

In this section, we present the Adaptive Spectral Partition (ASP) algorithm, whose pseudo-code is given in Algorithm 1, and prove that it mis-classifies less than $s = o(n)$ w.h.p. whenever this is at all possible, i.e., when (1) holds.

## 4.1 Algorithm and its optimality

The design of the ASP algorithm leverages the results derived to establish fundamental performance limits. In particular, we know that an optimal sampling strategy corresponds to $\boldsymbol{x}^*(\boldsymbol{p}, \boldsymbol{\alpha}) = \arg\max_{\boldsymbol{x} \in \mathcal{X}(\alpha)} \Delta(\boldsymbol{x}, \boldsymbol{p})$. ASP consists in three main steps: (i) first, we use a small fraction of the observation budget and apply spectral methods to obtain initial cluster estimates (Line 1 in Algorithm 1); (ii) the latter are then used to derive precise estimators of the SBM parameters, which in turn yield an estimate $\hat{\boldsymbol{x}}^*$ of $\boldsymbol{x}^*(\boldsymbol{p}, \boldsymbol{\alpha})$ (Lines 2 and 3); (iii) finally, $\hat{\boldsymbol{x}}^*$ dictates the way to sample edges with the remaining budget, and based on these additional observations, the cluster estimates are improved (Lines 4 and 5). The complexity of the ASP is polynomial to both $n$ and $T$. Indeed, Step 1, including the Spectral Clustering Algorithm, requires $O(T \log(n))$ operations. Step 2 requires $O(T)$ operations to estimate parameters and Step 3 solves a linear program where the number of variables is $k^2$ which does not scale with $n$ and $T$. The remaining steps simply check the log-likelihood values of each node, which requires $O(T)$ computations. Overall, the computational complexity of ASP is $O(T \log n)$.

We analyze the performance of ASP under the following mild assumptions, essentially stating some kind of homogeneity of the SBM parameters associated to the various clusters. There exist positive constants $\kappa_L$ and $\kappa_U$ such that

$$
\text{(A1)} \quad \left| \log \left( \frac{p_{ik}(1 - p_{jk})}{p_{jk}(1 - p_{ik})} \right) \right| \leq \kappa_U \quad \text{for all} \quad i, j, k
$$

$$
\text{(A2)} \quad \kappa_L \leq \left| \log \left( \frac{p_{ik}}{p_{jk}} \right) \right| \quad \text{for all} \quad i, j, k.
$$

We emphasize that no other assumptions are made on $\boldsymbol{p}$.

**Theorem 2.** *Assume that (A1) and (A2) hold. Let $s = o(n)$. The ASP algorithm mis-classifies less than $s$ nodes with high probability, if $\liminf_{n \to \infty} \frac{2TD(\boldsymbol{p}, \boldsymbol{\alpha})}{n \log(n/s)} \geq 1$.*

The above theorem is proved in the next subsection. In addition, the following lemma, proved in Appendix, directly implies that the ASP algorithm is $(s, \beta)$-locally stable at $(\boldsymbol{p}, \boldsymbol{\alpha})$ for $\beta = \frac{s}{n} \log(\frac{n}{s})$ when (1) holds.

**Lemma 1.** *Assume that (A1) and (A2) hold. For all $\boldsymbol{\alpha}$ and $\tilde{\boldsymbol{\alpha}}$, $\frac{|D(\boldsymbol{p}, \boldsymbol{\alpha}) - D(\boldsymbol{p}, \tilde{\boldsymbol{\alpha}})|}{D(\boldsymbol{p}, \boldsymbol{\alpha}) \|\tilde{\boldsymbol{\alpha}} - \boldsymbol{\alpha}\|_2} = O(1).$*

---
**Algorithm 1** Adaptive Spectral Partition$(T, \delta, K)$

---
**1. Initial random observations.**

Sample $\frac{T}{4\log(T/n)}$ edges uniformly at random without replacement and compute $\delta \leftarrow \frac{1}{\log\frac{e(\mathcal{V},\mathcal{V})T}{d(\mathcal{V},\mathcal{V})n}}$

Sample $\frac{\delta T}{4} - \frac{T}{4\log(T/n)}$ additional edges uniformly at random without replacement

Extract $\mathcal{S}_1, \ldots, \mathcal{S}_K$ using the spectral clustering algorithm of [16]

**2. Estimating the SBM parameters.** Estimate $\hat{\boldsymbol{\alpha}}$ and $\hat{\boldsymbol{p}}$ from the observations made in 1. and the extracted clusters $\mathcal{S}_1, \ldots, \mathcal{S}_K$:

$\hat{\alpha}_i = \frac{|\mathcal{S}_i|}{n}$ and $\hat{p}_{ii} = \frac{4e(\mathcal{S}_i, \mathcal{S}_i)}{\delta T} \frac{n(n-1)}{|\mathcal{S}_i|(|\mathcal{S}_i|-1)}$ for all $1 \le i \le K$ and $\hat{p}_{ij} = \frac{4e(\mathcal{S}_i, \mathcal{S}_j)}{\delta T} \frac{n(n-1)}{2|\mathcal{S}_i||\mathcal{S}_j|}$ for all $i \ne j$.

**3. Computing the optimal sampling strategy.** Solve $\hat{\boldsymbol{x}}^* \in \arg\max_{\boldsymbol{x} \in \mathcal{X}(\hat{\boldsymbol{\alpha}})} D(\boldsymbol{x}, \hat{\boldsymbol{p}})$

**4. First round of cluster improvement.**

$\hat{p} \leftarrow \max_{i,j} \hat{p}_{ij}$; $e(A, B) \leftarrow 0$ for all $A$, $B$ (i.e., reset all pairs)

Randomly observe $2(1 - \frac{\delta}{2})\hat{x}_{ij}^* \frac{T}{n}$ edges between node $v \in \mathcal{S}_i$ and nodes in $\mathcal{S}_j$ for all $v \in \mathcal{S}_i$ and for all $1 \le i, j \le K$

**for all** $1 \le i \le K$ **do**
$\qquad \hat{\mathcal{V}}_i = \emptyset$
$\qquad$ **for all** $v \in \mathcal{S}_i$ **do**
$\qquad\qquad$ Add $v$ to $\hat{\mathcal{V}}_i$ when $\max_{1 \le k \le K} \left| e(v, \mathcal{S}_k) - 2(1 - \frac{\delta}{2})\hat{x}_{ik}^* \hat{p}_{ik} \frac{T}{n} \right| \le \frac{\delta}{4} \hat{p} \frac{T}{n}$
$\qquad$ **end for**
**end for**

**5. Second round of cluster improvement.**

$e(A, B) \leftarrow 0$ for all $A$, $B$ (i.e., reset all pairs)

**for all** $v \in \cup_{k=1}^{K}(\mathcal{S}_k \setminus \hat{\mathcal{V}}_k)$ **do**
$\qquad$ Randomly select $\frac{\delta}{4K} \frac{T}{n - \sum_{k=1}^{K}|\hat{\mathcal{V}}_k|}$ nodes from $\hat{\mathcal{V}}_i$ for all $i$; and observe the edges between $v$ and the selected nodes
$\qquad v$ is assigned to $\hat{\mathcal{V}}_{k^*}$ where
$\qquad k^* = \arg\max_{1 \le i \le K} \sum_{k=1}^{K} \left( e(v, \mathcal{S}_k) \log(\hat{p}_{ik}) + (\frac{\delta}{4K} \frac{T}{n - \sum_{k=1}^{K}|\hat{\mathcal{V}}_k|} - e(v, \mathcal{S}_k)) \log(1 - \hat{p}_{ik}) \right).$
**end for**

---

## 4.2 The steps of ASP and their analysis

We describe below the various steps of the ASP algorithm, and analyze their performance. The proofs of all lemmas are postponed to the Appendix. In Algorithm 1, the pseudo-code of ASP, $e(\mathcal{A}, \mathcal{B})$ denotes the number of positive observations between nodes in $\mathcal{A}$ and nodes in $\mathcal{B}$, and $d(\mathcal{A}, \mathcal{B})$ denotes the number of sampled edges between $\mathcal{A}$ and $\mathcal{B}$.

### 4.2.1 Initial random observations

The first task of the algorithm is to collect $\frac{\delta T}{4}$ random edge samples so as to build initial cluster estimates and approximate the SBM parameters $(\boldsymbol{p}, \boldsymbol{\alpha})$ which in turn will lead to an approximate optimal sampling strategy $x^*(\boldsymbol{p}, \boldsymbol{\alpha})$. To output initial cluster estimates, we plan to use the spectral decomposition algorithm from [16], and to exploit the results therein about its performance. With this goal in mind, the parameter $\delta$ must be set so that the initial cluster estimates have an appropriate level of accuracy. To set $\delta$, we first estimate $p = \sum_{i=1}^{K} \sum_{j=1}^{K} \alpha_i \alpha_j p_{ij}$ from the first $\frac{T}{4\log(T/n)}$ samples by $\frac{e(\mathcal{V},\mathcal{V})}{d(\mathcal{V},\mathcal{V})}$. We then let $\delta = \left( \log \frac{e(\mathcal{V},\mathcal{V})T}{d(\mathcal{V},\mathcal{V})n} \right)^{-1}$ which is approximately equal to $\left( \log \frac{pT}{n} \right)^{-1}$. The spectral decomposition algorithm outputs $\mathcal{S}_1, \ldots, \mathcal{S}_K$, which in view of Theorem 2 in [16], satisfy with high probability, and for some constant $C > 0$,

$$\left| \cup_{i=1}^{K} \mathcal{V}_i \setminus \mathcal{S}_i \right| \le n \exp\left( -C \frac{pT/n}{\log(pT/n)} \right). \tag{2}$$

### 4.2.2 Estimating the SBM parameters

Next, using the initial cluster estimates, ASP approximates the SBM parameters $(\boldsymbol{p}, \boldsymbol{\alpha})$ by $(\hat{\boldsymbol{p}}, \hat{\boldsymbol{\alpha}})$ (refer to Line 2 in Algorithm 1). The previous step extracted the hidden clusters with at most

$n \exp(-\frac{T}{n}Cp)$ mis-classified nodes. This observation directly implies that: for any $k$,

$$|\hat{\alpha}_k - \alpha_k| \le \exp\left(-C\frac{pT/n}{\log(pT/n)}\right).$$

In addition, we can show that the initial cluster estimates also lead to a very accurate estimate of $\boldsymbol{p}$, as stated in the following lemma.

**Lemma 2.** *Assume that (A1) and (A2) hold. When $\left|\cup_{i=1}^{K}\mathcal{V}_i \setminus \mathcal{S}_i\right| \le n \exp\left(-C\frac{pT/n}{\log(pT/n)}\right)$ and $\frac{pT}{n} = \omega(1)$, with high probability, $\frac{|p_{ij} - \hat{p}_{ij}|}{p_{ij}} = O\left(\frac{\log(Tp/n)}{Tp/n} + \frac{1}{\sqrt{n}}\right).$*

### 4.2.3 Computing the optimal sampling strategy

ASP now computes $\hat{\boldsymbol{x}}^* \in \arg\max_{\boldsymbol{x} \in \mathcal{X}(\alpha)} \Delta(\hat{\boldsymbol{x}}, \hat{\boldsymbol{\alpha}})$. The main idea behind ASP is to use $\hat{\boldsymbol{x}}^*$ to define which edges should be sampled. However, $\hat{\boldsymbol{x}}^*$ defines how many times edges from cluster $\mathcal{V}_i$ to cluster $\mathcal{V}_j$ should be sampled, and these clusters are for now just approximated by $\mathcal{S}_i$ and $\mathcal{S}_j$. Hence using $\hat{\boldsymbol{x}}^*$ induces two sources of errors: first $\hat{\boldsymbol{x}}^*$ is inexact and then when randomly sampling an edge from $v \in \mathcal{V}_i$ to $\mathcal{S}_j$, the binary outcome is a Bernoulli r.v. with mean $\bar{p}_{ij}$ rather than $p_{ij}$, where

$$\bar{p}_{ij} = \frac{1}{|\mathcal{S}_j|} \sum_{k=1}^{K} p_{ik}|\mathcal{S}_j \cap \mathcal{V}_k|.$$

Let $\bar{\boldsymbol{p}} = [\bar{p}_{ij}]_{i,j}$. The following lemma is instrumental to bound the impact of these errors:

**Lemma 3.** *Assume that (A1) and (A2) hold. When $\left|\cup_{i=1}^{K}\mathcal{V}_i \setminus \mathcal{S}_i\right| \le n \exp\left(-C\frac{pT/n}{\log(pT/n)}\right)$, $\frac{|p_{ij} - \hat{p}_{ij}|}{p_{ij}} = O\left(\frac{\log(Tp/n)}{Tp/n} + \frac{1}{\sqrt{n}}\right)$, and $\frac{pT}{n} = \omega(1)$, with high probability,*

$$\frac{|\Delta(\boldsymbol{x}^*(\hat{\boldsymbol{p}}, \hat{\boldsymbol{\alpha}}), \bar{\boldsymbol{p}}) - \Delta(\boldsymbol{x}^*(\boldsymbol{p}, \boldsymbol{\alpha}), \boldsymbol{p})|}{\Delta(\boldsymbol{x}^*(\boldsymbol{p}, \boldsymbol{\alpha}), \boldsymbol{p})} = O(\frac{\log(Tp/n)}{Tp/n} + \frac{1}{\sqrt{n}}),$$

### 4.2.4 First round of cluster improvement

In this step, the ASP algorithm first re-sets the values of $e(\mathcal{A}, \mathcal{B})$ to 0 for all sets of nodes $\mathcal{A}$ and $\mathcal{B}$. It then randomly samples edges according to the sampling strategy $\hat{\boldsymbol{x}}^*$. More precisely, for every $v \in \mathcal{S}_i$, for each $k = 1, \dots, K$, it randomly selects $2(1 - \frac{\delta}{2})\hat{x}_{ik}^* \frac{T}{n}$ edges from $v$ to $\mathcal{S}_k$. Due to the re-set, after these new samples, $e(v, \mathcal{S}_k)$ is a sum of independent Bernoulli r.v. with mean $\bar{p}_{jk}$ when $v \in \mathcal{V}_j$.

In this first round of cluster improvement, ASP classifies a node $v \in \mathcal{S}_i$ only if the values $e(v, \mathcal{S}_k)$'s clearly indicate its cluster. More precisely, $v \in \mathcal{S}_i$ is classified in $\hat{\mathcal{V}}_i$ only if:

$$\max_{1 \le k \le K} \left| e(v, \mathcal{S}_k) - 2(1 - \frac{\delta}{2})\hat{x}_{ik}^*\hat{p}_{ik}\frac{T}{n} \right| \le \frac{\delta}{4}\hat{p}\frac{T}{n}. \tag{3}$$

Note that $\mathbb{E}[e(v, \mathcal{S}_k)] = 2(1 - \frac{\delta}{2})\hat{x}_{ik}^*\bar{p}_{ik}\frac{T}{n}$ when $v \in \mathcal{S}_i \cap \mathcal{V}_i$. When $v \in \mathcal{S}_i \cap \mathcal{V}_i$, $v$ satisfies (3) with probability at least $1 - \exp(-\frac{pT/n}{(\log(pT/n))^3})$ from Chernoff-Hoeffding inequality. Therefore, from (2) and the Markov inequality, with high probability,

$$\left|\mathcal{S}_i \setminus \hat{\mathcal{V}}_i\right| \le |\mathcal{S}_i \setminus \mathcal{V}_i| + \left|(\mathcal{V}_i \cap \mathcal{S}_i) \setminus \hat{\mathcal{V}}_i\right| \le \alpha_i n e^{-\frac{pT/n}{(\log(pT/n))^4}}.$$

All the nodes in $\cup_{k=1}^{K}(\mathcal{S}_i \setminus \hat{\mathcal{V}}_i)$ will be classified in the second round of cluster improvement.

When $v \in \mathcal{S}_i \cap \mathcal{V}_j$ for $j \ne i$, $\mathbb{E}[e(v, \mathcal{S}_k)] = 2(1 - \frac{\delta}{2})\hat{x}_{ik}^*\bar{p}_{jk}\frac{T}{n}$ which could be far from $2(1 - \frac{\delta}{2})\hat{x}_{ik}^*\hat{p}_{ik}\frac{T}{n}$. Thus, to satisfy (3), $e(v, \mathcal{S}_k)$ has to deviate from its mean a lot. From Chernoff-Hoeffding inequality, we show that (3) holds with probability at most $\exp(-\frac{2T}{n}(\Delta(\boldsymbol{x}^*(\hat{\boldsymbol{p}}, \hat{\boldsymbol{\alpha}}), \bar{\boldsymbol{p}}) + O(\delta p)))$. Thus, using Lemma 3 and the Markov inequality, with high probability, $\left|\hat{\mathcal{V}}_i \setminus \mathcal{V}_i\right| \le \alpha_i n e^{-2\frac{T}{n} \cdot D(\boldsymbol{p}, \boldsymbol{\alpha}) + O(\frac{\delta pT}{n})}$.

In summary, we have:

**Lemma 4.** *Assume that (A1) and (A2) hold. After the first round of cluster improvement of the ASP algorithm, we have, with high probability, for all $1 \le i \le K$,*

$$\left| \mathcal{S}_i \setminus \hat{\mathcal{V}}_i \right| \le \alpha_i n e^{-\frac{pT/n}{(\log(pT/n))^4}} \quad and \quad \left| \hat{\mathcal{V}}_i \setminus \mathcal{V}_i \right| \le \alpha_i n e^{-2\frac{T}{n} \cdot D(\boldsymbol{p}, \boldsymbol{\alpha}) + O\left(\frac{\delta pT}{n}\right)}.$$

### 4.2.5 Second round of cluster improvement

Finally, we use the $\frac{\delta}{4}T$ remaining samples to classify the nodes in $\cup_{k=1}^{K}(\mathcal{S}_i \setminus \hat{\mathcal{V}}_i)$ that were not assigned to a cluster in the previous round. From Lemma 4, there are at most $n \exp\left(-\frac{pT/n}{(\log(pT/n))^4}\right)$ such nodes. Hence, we can use at least $\frac{\delta T}{4n} \exp\left(\frac{pT/n}{(\log(pT/n))^4}\right)$ samples to classify each remaining node.

In the second round of cluster improvement, the ASP algorithm classifies nodes using the maximum likelihood with $\hat{\boldsymbol{p}}$, which is very similar to the greedy improvement step of the spectral clustering algorithm of [16]. Define:

$$D^R(\boldsymbol{p}, \boldsymbol{\alpha}) = \min_{i,j:i \neq j} D_+(p_i, p_j, \boldsymbol{\alpha}) \quad \text{with}$$

$$D_+(p_i, p_j, \boldsymbol{\alpha}) = \min_{y \in [0,1]^K} \max \left\{ \sum_{k=1}^{K} \alpha_k KL(y_k, p_{ik}), \sum_{k=1}^{K} \alpha_k KL(y_k, p_{jk}) \right\}.$$

From the definition of $D^R(\hat{\boldsymbol{p}}, \hat{\boldsymbol{\alpha}})$, we can show as in [16] that $v$ is mis-classified only when

$$\sum_{k=1}^{K} d(v, \mathcal{S}_k) KL\left(\frac{e(v, \mathcal{S}_k)}{d(v, \mathcal{S}_k)}, \hat{p}_{ik}\right) \ge d(v, \mathcal{V}) D^R(\hat{\boldsymbol{p}}, \hat{\boldsymbol{\alpha}}).$$

From Chernoff-Hoeffding inequality, we analyze the probability of the above event and conclude:

**Lemma 5.** *The second round of cluster improvement generates at most* $n \exp\left(-\frac{pT}{n} \exp\left(\frac{pT/n}{(\log(pT/n))^5}\right)\right)$ *mis-classified nodes with high probability under (A1) and (A2).*

So far we have analyzed the number of mis-classified nodes generated in the first round and the second round of cluster improvement. In the first round, the ASP algorithm generates at most $ne^{-2\frac{T}{n} \cdot D(\boldsymbol{p}, \boldsymbol{\alpha}) + O\left(\frac{\delta pT}{n}\right)}$ mis-classified nodes and in the second round, generates at most $ne^{-\frac{pT}{n} \exp\left(\frac{pT/n}{(\log(pT/n))^5}\right)}$ mis-classified nodes. Note that the number of mis-classified nodes in the second round is negligible compared to that in the first round. We thus conclude that the ASP algorithm outputs cluster estimates with at most

$$\left| \cup_{i=1}^{K} \hat{\mathcal{V}}_i \setminus \mathcal{V}_i \right| \le n e^{-2\frac{T}{n} \cdot D(\boldsymbol{p}, \boldsymbol{\alpha}) + O\left(\frac{\delta pT}{n}\right)}$$

mis-classified nodes, which concludes the proof of Theorem 2, in view of our choice of $\delta$.

## 5 Conclusion

In this paper, we derived a necessary condition for the existence of a joint sampling and clustering algorithm mis-classifying less than $s = o(n)$ nodes w.h.p. in the SBM. This derivation revealed the optimal sampling strategy, and allowed us to devise ASP, an algorithm that mis-classified $s = o(n)$ when the necessary condition holds. To our knowledge, this is the first time the optimal cluster recovery rate is characterized in the SBM with adaptive sampling. Being able to characterize the optimal sampling strategy is promising, and opens up new research directions. In particular, we could now investigate various online optimization problems involving random graphs generated by the SBM and using adaptive edge sampling.

### Acknowledgments

S. Yun was supported by Korea Electric Power Corporation. (Grant number:R18XA05). A. Proutiere was partially supported by the Wallenberg Autonomous Systems and Software Program (WASP) funded by the Knut and Alice Wallenberg Foundation.

## Footnotes

[1] w.h.p. means that the probability tends to 1 as $n$ grows large.

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
