[Supplementary Material]

# Optimal Sampling and Clustering
# in the Stochastic Block Model
# (Supplementary material)

## A   Proof of Theorem 1 – Binary symmetric SBMs

Consider the binary symmetric SBM, where $\boldsymbol{\alpha} = (\frac{1}{2}, \frac{1}{2})$, $p_{11} = p_{22} = p$, and $p_{12} = p_{21} = q$. Without loss of generality, let $\mathcal{V}_1 = \{1, \ldots, \frac{n}{2}\}$ and $\mathcal{V}_2 = \{\frac{n}{2} + 1, \ldots, n\}$. Assume that there exists a joint sampling and clustering algorithm $\pi$ satisfying[1]:

$$\mathbb{P}\big[\big|\cup_{k=1}^2 \mathcal{V}_k \setminus \mathcal{S}_k^\pi\big| < s\big] \geq 1 - \eta_n, \tag{1}$$

where $\mathcal{S}_1^\pi, \mathcal{S}_2^\pi$ are the clusters returned by the algorithm $\pi$ and $\lim_{n\to\infty} \eta_n = 0$. We show that (1) cannot hold when

$$\frac{2T}{n} \max\{KL(p,q), KL(q,p)\} \leq \frac{1}{(1+\gamma_n)^4} \log\left(\frac{n}{s}\right), \tag{2}$$

where $\gamma_n = \left(\log\left(\frac{n}{s}\right)\right)^{-1/4} + \sqrt{\eta_n}$. Note that $\lim_{n\to\infty} \gamma_n = 0$. To this aim, we first relate (1) to an hypothesis testing problem, where each hypothesis corresponds to an allocation of nodes to clusters. We then construct two stochastic models: the first model leads to observations generated under the true SBM, and the second to observations made under a different allocation of nodes to clusters. Finally, we use a change-of-measure argument: we study the log-likelihood ratio of the observations under the two models, and show how it relates to hypothesis testing problem and to the error probability $\eta_n$. This leads to our necessary condition.

**Hypothesis testing.** Let $\bar{s} = \lceil\frac{s+1}{\gamma_n}\rceil$, consider the hypotheses $H_0, \ldots, H_M$ corresponding to different allocations of nodes to clusters. Let $\mathcal{V}_1^{(m)}, \mathcal{V}_2^{(m)}$ be the allocation corresponding to $H_m$. We construct these allocations so that:

(C0)   $\mathcal{V}_1^{(0)} = \{1, 2, \ldots, \frac{n}{2}\}$   and   $\mathcal{V}_2^{(0)} = \{\frac{n}{2} + 1, \ldots, n\}$,

(C1)   $\left|\cup_{k=1}^2 \mathcal{V}_k^{(0)} \setminus \mathcal{V}_k^{(m)}\right| = 2\bar{s}$   for all   $1 \leq m \leq M$,

(C2)   $\left|\cup_{k=1}^2 \mathcal{V}_k^{(m)} \setminus \mathcal{V}_k^{(l)}\right| \geq 2\bar{s}$   for all   $m, l : m \neq l$.

We can prove that we can build $M \geq \left(\frac{n}{32e\bar{s}}\right)^{\bar{s}}$ hypotheses satisfying $(C1)$ and $(C2)$. This lower bound on $M$ comes from the analysis of $\frac{\binom{n/2}{\bar{s}}^2}{\sum_{l=1}^{\bar{s}} \binom{2\bar{s}}{l}\binom{n-2\bar{s}}{l}}$: there are $\binom{n/2}{\bar{s}}^2$ feasible allocations satisfying $(C1)$ and for any such given allocation, there are at most $\sum_{l=1}^{\bar{s}} \binom{2\bar{s}}{l}\binom{n-2\bar{s}}{l}$ allocations violating $(C2)$. A detailed proof is provided in (18).

Based on the outputs of the algorithm $\pi$, define the following hypothesis testing function:

$$f(\mathcal{S}_1^\pi, \mathcal{S}_2^\pi) = \arg\min_{m \in \{0, 1, \ldots, M\}} \left|\cup_{k=1}^2 \mathcal{V}_k^{(m)} \setminus \mathcal{S}_k^\pi\right|.$$

Note that $\left|\cup_{k=1}^2 \mathcal{V}_k^{(m)} \setminus \mathcal{V}_k^{(l)}\right|$ can be interpreted as a distance between any two different allocations $(\mathcal{V}_1^{(m)}, \mathcal{V}_2^{(m)})$ and $(\mathcal{V}_1^{(l)}, \mathcal{V}_2^{(l)})$ and thus the triangle inequality holds, e.g.,

$$\left|\cup_{k=1}^2 \mathcal{V}_k^{(m)} \setminus \mathcal{V}_k^{(l)}\right| \leq \left|\cup_{k=1}^2 \mathcal{S}_k^\pi \setminus \mathcal{V}_k^{(m)}\right| + \left|\cup_{k=1}^2 \mathcal{S}_k^\pi \setminus \mathcal{V}_k^{(l)}\right| \quad \text{for all} \quad m, l.$$

We deduce from (1) and the triangle inequality that:

$$\mathbb{P}[f(\mathcal{S}_1^\pi, \mathcal{S}_2^\pi) = 0] \geq 1 - \eta_n \quad \text{and} \quad \min_{1 \leq m \leq M} \mathbb{P}[f(\mathcal{S}_1^\pi, \mathcal{S}_2^\pi) = m] \leq \frac{\eta_n}{M}, \tag{3}$$

since the algorithm $\pi$ mis-classifies at most $s$ nodes with probability $1 - \eta_n$.

**Two stochastic models.** To prove the necessary condition, we use a change-of-measure argument. More precisely, we consider that the observations can either come from the true stochastic model $\Phi$ corresponding to the true allocation of nodes to clusters, or from another model $\Psi$. Let $\mathbb{P}_\Phi = \mathbb{P}$ (resp. $\mathbb{P}_\Psi$) and $\mathbb{E}_\Phi[\cdot] = \mathbb{E}[\cdot]$ (resp. $\mathbb{E}_\Psi[\cdot]$) be the probability measure and expectation under the model $\Phi$ (resp. $\Psi$). To construct the second model $\Psi$, let $m^* = \arg\min_{1 \leq m \leq M} \mathbb{P}[f(\mathcal{S}_1^\pi, \mathcal{S}_2^\pi) = m]$. $\Psi$ is constructed so that:

$$\mathbb{P}_\Psi[f(\mathcal{S}_1^\pi, \mathcal{S}_2^\pi) = m^*] \geq 1 - \eta_n. \tag{4}$$

The construction of such a $\Psi$ can be made as follows: randomly select $\frac{\bar{s}+s+1}{2}$ nodes from $\mathcal{V}_1 \setminus \mathcal{V}_1^{(m^*)}$ and $\frac{\bar{s}+s+1}{2}$ nodes from $\mathcal{V}_2 \setminus \mathcal{V}_2^{(m^*)}$ and swap the selected nodes. We denote by $\tilde{\mathcal{V}}_1$ and $\tilde{\mathcal{V}}_2$ the resulting clusters under $\Psi$. Then,

$$\left|\cup_{k=1}^2 \tilde{\mathcal{V}}_k \setminus \mathcal{V}_k^{(m^*)}\right| = \bar{s} - s - 1 \quad \text{and} \quad \left|\cup_{k=1}^2 \tilde{\mathcal{V}}_k \setminus \mathcal{V}_k^{(0)}\right| = \bar{s} + s + 1. \tag{5}$$

From the triangle inequality and the condition $(C2)$, for all $m \neq m^*$

$$\left|\cup_{k=1}^2 \mathcal{V}_k^{(m)} \setminus \tilde{\mathcal{V}}_k\right| \geq \left|\cup_{k=1}^2 \mathcal{V}_k^{(m)} \setminus \mathcal{V}_k^{(m^*)}\right| - \left|\cup_{k=1}^2 \tilde{\mathcal{V}}_k \setminus \mathcal{V}_k^{(m^*)}\right| \geq \bar{s} + s + 1. \tag{6}$$

Now, since $\pi$ must have by assumption less than $s$ mis-classified nodes with probability $1 - \eta_n$, we deduce, combining (5) and (6), that (4) actually holds.

**Change-of-measure argument.** In (3) and (4), we have identified events that are very unlikely under $\mathbb{P}_\Phi$ and very likely under $\mathbb{P}_\Psi$ when we assume that the algorithm mis-classified less than $s$ nodes w.h.p.. More precisely, we deduce from (3) and (4) that:

$$\mathbb{P}_\Phi[f(\mathcal{S}_1^\pi, \mathcal{S}_2^\pi) = m^*] \leq \frac{\eta_n}{M} \quad \text{and} \quad \mathbb{P}_\Psi[f(\mathcal{S}_1^\pi, \mathcal{S}_2^\pi) = m^*] \geq 1 - \eta_n. \tag{7}$$

This observation will be used to derive an upper bound of the log-likelihood ratio $\mathcal{Q}$ of the observations under $\mathbb{P}_\Phi$ and $\mathbb{P}_\Psi$. Let $y(t)$ and $e(t)$ denote the value and the edge corresponding to the $t$-th observation, respectively. Let $\mathcal{C}_{ij} = \{(v, w) \in (\tilde{\mathcal{V}}_i \setminus \mathcal{V}_i^{(0)}) \times (\tilde{\mathcal{V}}_j \cap \mathcal{V}_j^{(0)})\}$. Then, the log-likelihood ratio of the observations under $\mathbb{P}_\Phi$ and $\mathbb{P}_\Psi$, $\mathcal{Q}$, is defined as follows:

$$\mathcal{Q} = \sum_{t=1}^T \mathbb{1}(e(t) \in \mathcal{C}_{11} \cup \mathcal{C}_{22}) \left(y(t) \log \frac{p}{q} + (1 - y(t)) \log \frac{1-p}{1-q}\right) +$$
$$\sum_{t=1}^T \mathbb{1}(e(t) \in \mathcal{C}_{12} \cup \mathcal{C}_{21}) \left(y(t) \log \frac{q}{p} + (1 - y(t)) \log \frac{1-q}{1-p}\right).$$

Using (7), we can upper bound $\mathbb{P}_\Psi\{\mathcal{Q} \leq \log(M)\}$ as follows. Observe first that:

$$\begin{aligned}
\mathbb{P}_\Psi\{\mathcal{Q} \leq \log(M), f(\mathcal{S}_1^\pi, \mathcal{S}_2^\pi) = m^*\} &= \int_{\{\mathcal{Q} \leq \log(M), f(\mathcal{S}_1^\pi, \mathcal{S}_2^\pi) = m^*\}} d\mathbb{P}_\Psi \\
&= \int_{\{\mathcal{Q} \leq \log(M), f(\mathcal{S}_1^\pi, \mathcal{S}_2^\pi) = m^*\}} \exp(\mathcal{Q}) d\mathbb{P}_\Phi \\
&\leq M \mathbb{P}_\Phi\{\mathcal{Q} \leq \log(M), f(\mathcal{S}_1^\pi, \mathcal{S}_2^\pi) = m^*\} \\
&\leq M \mathbb{P}_\Phi\{f(\mathcal{S}_1^\pi, \mathcal{S}_2^\pi) = m^*\} \\
&\leq \eta_n. 
\end{aligned} \tag{8}$$

From (7), we also have:

$$\mathbb{P}_\Psi\{\mathcal{Q} \le \log(M), f(\mathcal{S}_{i*}^\pi) \ne m^*\} \le \mathbb{P}_\Psi\{f(\mathcal{S}_{i*}^\pi) \ne m^*\} \le \eta_n. \tag{9}$$

Combining (8) and (9), we conclude that:

$$\mathbb{P}_\Psi\{\mathcal{Q} > \log(M)\} \ge 1 - 2\eta_n. \tag{10}$$

Now since nodes within the same cluster play the same role and due to the symmetry of the SBM, the expected number of observations should be identical for all nodes. Thus:

$$\mathbb{E}_\Psi\left[\sum_{t=1}^T \mathbb{1}(e(t) \in \cup_{i,j}\mathcal{C}_{ij})\right] \le \frac{2(\bar{s} + s + 1)}{n}T. \tag{11}$$

Moreover, we have:

$$\mathbb{E}_\Psi\left[y(t)\log\frac{p}{q} + (1 - y(t))\log\frac{1-p}{1-q}|e(t) \in \mathcal{C}_{11} \cup \mathcal{C}_{22}\right] = KL(p, q)$$

$$\mathbb{E}_\Psi\left[y(t)\log\frac{q}{p} + (1 - y(t))\log\frac{1-q}{1-p}|e(t) \in \mathcal{C}_{12} \cup \mathcal{C}_{21}\right] = KL(q, p),$$

and from the optional stopping time theorem and (11),

$$\mathbb{E}_\Psi[\mathcal{Q}] \le (\bar{s} + s + 1)\frac{2T}{n}\max\{KL(p, q), KL(q, p)\}. \tag{12}$$

In the above inequality, $\mathbb{E}_\Psi[\mathcal{Q}]$ is maximized when all the observation budget is used to sample inter-cluster edges when $KL(p, q) < KL(q, p)$ and intra-cluster edges when $KL(p, q) > KL(q, p)$ (which corresponds to $\boldsymbol{x}^*(\boldsymbol{p}, \boldsymbol{\alpha})$). Markov inequality then implies that:

$$\mathbb{P}_\Psi\{\mathcal{Q} > \log(M)\} \le \frac{2T(\bar{s} + s + 1)\max\{KL(p, q), KL(q, p)\}}{n\log(M)}. \tag{13}$$

However, in view of (2) and of the definitions of $\bar{s}$ and $M$, $\frac{2T(\bar{s}+s+1)\max\{KL(p,q),KL(q,p)\}}{n\log(M)}$ cannot exceed $1 - 2\eta_n$. Hence (13) contradicts (10). Therefore, to have less than $s$ mis-classified nodes with high probability, we need: $\liminf_{n\to\infty} \frac{2T\cdot\max\{KL(p,q),KL(q,p)\}}{n\log(n/s)} \ge 1$.

## B Proof of Theorem 1 – Generic SBMs

**Notations.** Let $\mathbb{1}_S \in \{0, 1\}^n$ be the binary vector where only the values corresponding to the indices of elements in $S$ are equal to 1. Let $d^H(A, B)$ denote the Hamming distance between $A$ and $B$. We denote by $\mathbb{P}^\pi$ and $\mathbb{E}^\pi$ the probability and the expectation under a given sampling and clustering algorithm $\pi$.

**Proof strategy.** Assume that an algorithm $\pi$ satisfies:

(P1) when $|\mathcal{V}_i| = \alpha_i n$ for all $1 \le i \le K$,

$$\mathbb{E}^\pi[d(v, \mathcal{V}_j)] = 2x_{ij}^\pi\frac{T}{n} \quad \text{for all} \quad v \in \mathcal{V}_i \quad \text{for all} \quad 1 \le i, j \le K, \tag{14}$$

(P2) for all $(\mathcal{V}_1, \ldots, \mathcal{V}_K)$ having $\sqrt{\sum_{k=1}^K (\alpha_k - \frac{|\mathcal{V}_k|}{n})^2} \le \frac{s}{n}\log(\frac{n}{s}) \le \beta$,

$$\mathbb{P}^\pi\left[\frac{1}{2}\sum_{k=1}^K d^H(\mathbb{1}_{\mathcal{V}_k}, \mathbb{1}_{\mathcal{S}_k^\pi}) < s\right] \ge 1 - \eta_n, \tag{15}$$

where $(\mathcal{S}_k^\pi)_{1\le k\le K}$ are the clusters returned by the algorithm $\pi$ and $\lim_{n\to\infty}\eta_n = 0$.

From $\eta_n$ and the target number of mis-classified nodes, we set $\gamma_n = \left(\log\left(\frac{n}{s}\right)\right)^{-1/4} + \sqrt{\eta_n}$ and $\bar{s} = \lceil \frac{s+1}{\gamma_n} \rceil$. In the proof, we show that with any algorithm $\pi$ and $\boldsymbol{x}^\pi \in \mathcal{X}(\alpha)$, it is not able to satisfy both (P1) and (P2) when

$$2\frac{T}{n} \cdot D(\boldsymbol{p}, \boldsymbol{\alpha}) < \frac{1}{(1+\gamma_n)^4} \log\left(\frac{n}{s}\right). \tag{16}$$

Let $i^*$ and $j^*$ be the cluster indices such that

$$\sum_{k=1}^K x_{i^*k}^\pi KL(p_{i^*k}, p_{j^*k}) = \Delta(\boldsymbol{x}^\pi, \boldsymbol{p}).$$

**Hypothesis testing for clusters $i^*$ and $j^*$.** Let $H_0, \dots, H_M$ be hypotheses defined with different partitions $(\mathcal{V}_1^{(m)}, \dots, \mathcal{V}_K^{(m)})$ satisfying that

(C0) $|\mathcal{V}_{i^*}^{(0)}| = \alpha_{i^*} n - \bar{s} - s - 1, \quad |\mathcal{V}_{j^*}^{(0)}| = \alpha_{j^*} n + \bar{s} + s + 1, \quad$ and

$\qquad |\mathcal{V}_k^{(0)}| = \alpha_k n \quad$ for all $\quad k \notin \{i^*, j^*\}$,

(C1) $\mathcal{V}_{i^*}^{(0)} \subset \mathcal{V}_{i^*}^{(m)} \quad$ and $\quad |\mathcal{V}_{i^*}^{(m)}| = \alpha_{i^*} n + \bar{s} - s - 1$,

(C2) $d^H(\mathbb{1}_{\mathcal{V}_{i^*}^{(m)}}, \mathbb{1}_{\mathcal{V}_{i^*}^{(l)}}) \geq 2\bar{s} \quad$ for all $\quad l \neq m$

(C3) $\mathcal{V}_k^{(m)} = \mathcal{V}_k^{(0)} \quad$ for all $\quad k \notin \{i^*, j^*\}$

Then, we can build this set of hypotheses such that

$$M \geq \left(\frac{\alpha_{j^*} n}{16 e \bar{s}}\right)^{\bar{s}}. \tag{17}$$

This is due to the following reason. There are $\binom{\alpha_{j^*} n + \bar{s} + s + 1}{2\bar{s}}$ different partitions satisfying (C0), (C1), and (C3), since we can generate a new partition by moving $2\bar{s}$ elements of $\mathcal{V}_{j^*}^{(0)}$ to $\mathcal{V}_{i^*}^{(m)}$. Let the first partition $(\mathcal{V}_1^{(1)}, \dots, \mathcal{V}_K^{(1)})$ be randomly selected among the $\binom{\alpha_{j^*} n + \bar{s} + s + 1}{2\bar{s}}$ possible partitions. For the given $(\mathcal{V}_1^{(1)}, \dots, \mathcal{V}_K^{(1)})$, there are at most $\sum_{l=1}^{\bar{s}} \binom{2\bar{s}}{l} \binom{\alpha_{j^*} n - \bar{s} + s + 1}{l}$ partitions satisfying (C0), (C1), and (C3), but not (C2). After removing all the partitions violating (C2) with respect to the first partition, we randomly select the second partition and then remove all the partitions violating (C2) with respect to the second partition. In this manner, we can iteratively generate $M$ partitions with

$$
\begin{aligned}
M &\geq \frac{\binom{\alpha_{j^*} n + \bar{s} + s + 1}{2\bar{s}}}{\sum_{l=1}^{\bar{s}} \binom{2\bar{s}}{l} \binom{\alpha_{j^*} n - \bar{s} + s + 1}{l}} \\
&\overset{(a)}{\geq} \frac{\left(\frac{\alpha_{j^*} n + \bar{s} + s + 1}{2\bar{s}}\right)^{2\bar{s}}}{\sum_{l=1}^{\bar{s}} \binom{2\bar{s}}{l} \binom{\alpha_{j^*} n - \bar{s} + s + 1}{l}} \\
&\geq \frac{\left(\frac{\alpha_{j^*} n}{2\bar{s}}\right)^{2\bar{s}}}{\sum_{l=0}^{\bar{s}} \binom{2\bar{s}}{l} \binom{\alpha_{j^*} n}{l}} \\
&\overset{(b)}{\geq} \frac{\left(\frac{\alpha_{j^*} n}{2\bar{s}}\right)^{2\bar{s}}}{2^{2\bar{s}} \left(\frac{e \alpha_{j^*} n}{\bar{s}}\right)^{\bar{s}}} \\
&= \left(\frac{\alpha_{j^*} n}{16 e \bar{s}}\right)^{\bar{s}}, \tag{18}
\end{aligned}
$$

where $(a)$ is obtained from $\binom{a}{b} \geq \left(\frac{a}{b}\right)^b$ and $(b)$ is obtained from the facts that $\sum_{l=0}^{\bar{s}} \binom{2\bar{s}}{l} < \sum_{l=0}^{2\bar{s}} \binom{2\bar{s}}{l} = 2^{2\bar{s}}$ and $\binom{a}{b} \leq \left(\frac{ea}{b}\right)^b$ for all $0 \leq b \leq a$.

**Random graph models.** Let $\Phi$ be the SBM model corresponding to the partition of $H_0$, $(\mathcal{V}_1^{(0)}, \dots, \mathcal{V}_K^{(0)})$, and $\Psi^{(m)}$ be the SBM model with partition $(\tilde{\mathcal{V}}_1^{(m)}, \cdots, \tilde{\mathcal{V}}_K^{(m)})$ such that

$$\mathcal{V}_{i^*}^{(0)} \subset \tilde{\mathcal{V}}_{i^*}^{(m)} \subset \mathcal{V}_{i^*}^{(m)} \quad \text{and} \quad |\tilde{\mathcal{V}}_{i^*}^{(m)}| = \alpha_{i^*} n. \tag{19}$$

Let $\mathbb{P}_\Phi$ and $\mathbb{P}_{\Psi^{(m)}}$ be the probability measures under the models $\Phi$ and $\Psi^{(m)}$, respectively. We analogously denote by $\mathbb{E}_\Phi$ and $\mathbb{E}_{\Psi^{(m)}}$ the expectations under $\Phi$ and $\Psi^{(m)}$.

From the clustering algorithm $\pi$, we can build a simple hypothesis test as follows:

$$f(\mathcal{S}_{i*}^\pi) = \arg \min_{m \in \{0,1,\ldots,M\}} d^H(\mathbb{1}_{\mathcal{V}_{i*}^{(m)}}, \mathbb{1}_{\mathcal{S}_{i*}^\pi}).$$

Note that the partition vector of $(\tilde{\mathcal{V}}_1^{(m)}, \ldots, \tilde{\mathcal{V}}_K^{(m)})$ is $\boldsymbol{\alpha}$ and the partition vector of $(\mathcal{V}_1^{(m)}, \ldots, \mathcal{V}_K^{(m)})$ is $\boldsymbol{\alpha}'$ satisfying $\|\boldsymbol{\alpha} - \boldsymbol{\alpha}'\|_2 = \frac{\bar{s}+s+1}{n}\sqrt{2} \le \frac{s}{n}\log\left(\frac{n}{s}\right)$. Thus, we have

$$\mathbb{P}_\Phi[f(\mathcal{S}_{i*}^\pi) = 0] \ge 1 - \eta_n, \quad \text{and} \quad \mathbb{P}_{\Psi^{(m)}}[f(\mathcal{S}_{i*}^\pi) = m] \ge 1 - \eta_n \quad \text{for all} \quad 1 \le m \le M, \quad (20)$$

since the number of mis-classified nodes is less than $s$ with probability at least $1 - \eta_n$ from (15); and the condition (C2) and the definition of $\Psi^{(m)}$ in (19) imply that $f(\mathcal{S}_i^\pi) = 0$ under $\Phi$ and $f(\mathcal{S}_i^\pi) = m$ under $\Psi^{(m)}$ when the number of mis-classified nodes is less than $s$.

**The log-likelihood ratio and its connection to the error probability of the hypothesis test.** Let $m^* = \arg \min_{m \in \{1,\ldots,M\}} \mathbb{P}_\Phi[f(\mathcal{S}_{i*}^\pi) = m]$. Then, from (20),

$$\mathbb{P}_\Phi[f(\mathcal{S}_{i*}^\pi) = m^*] \le \frac{\eta_n}{M}. \quad (21)$$

In what follows, we derive the log-likelihood ratio of the observations between $\Phi$ and $\Psi^{(m^*)}$ and explain its connection to the error probability.

Let $(\tilde{\mathcal{V}}_1, \ldots, \tilde{\mathcal{V}}_K)$ be the partition under $\Psi^{(m^*)}$. Let $y(t)$ and $e(t)$ denote the observed value and the observed edge at the $t$-th observation, respectively. Let $\mathcal{C}_k = \{(v, w) \in (\tilde{\mathcal{V}}_{i*} \setminus \mathcal{V}_{i*}^{(0)}) \times \tilde{\mathcal{V}}_k\}$. We introduce $\mathcal{Q}$, referred to as the pseudo-log-likelihood ratio of the observations under $\mathbb{P}_\Phi$ and $\mathbb{P}_{\Psi^{(m^*)}}$) as follows:

$$\mathcal{Q} = \sum_{t=1}^{T} \sum_{k=1}^{K} \mathbb{1}(e(t) \in \mathcal{C}_k) \left( y(t) \log \frac{p_{i*k}}{p_{j*k}} + (1 - y(t)) \log \frac{1 - p_{i*k}}{1 - p_{j*k}} \right). \quad (22)$$

We have:

$$\mathbb{P}_{\Psi^{(m^*)}}^\pi \{\mathcal{Q} \le \log(M)\}$$
$$= \mathbb{P}_{\Psi^{(m^*)}}^\pi \{\mathcal{Q} \le \log(M), f(\mathcal{S}_{i*}^\pi) = m^*\} + \mathbb{P}_{\Psi^{(m^*)}}^\pi \{\mathcal{Q} \le \log(M), f(\mathcal{S}_{i*}^\pi) \ne m^*\}. \quad (23)$$

We get:

$$\begin{aligned}
\mathbb{P}_{\Psi^{(m^*)}}^\pi \{\mathcal{Q} \le \log(M), f(\mathcal{S}_{i*}^\pi) = m^*\} &= \int_{\{\mathcal{Q} \le \log(M), f(\mathcal{S}_{i*}^\pi) = m^*\}} d\mathbb{P}_{\Psi^{(m^*)}}^\pi \\
&= \int_{\{\mathcal{Q} \le \log(M), f(\mathcal{S}_{i*}^\pi) = m^*\}} \exp(\mathcal{Q}) d\mathbb{P}_\Phi^\pi \\
&\le \exp(\log(M)) \mathbb{P}_\Phi^\pi \{\mathcal{Q} \le \log(M), f(\mathcal{S}_{i*}^\pi) = m^*\} \\
&\le M \mathbb{P}_\Phi^\pi \{f(\mathcal{S}_{i*}^\pi) = m^*\} \\
&\le \eta_n, \quad (24)
\end{aligned}$$

where the last inequality is obtained from (21).

From (20), we also have:

$$\begin{aligned}
\mathbb{P}_{\Psi^{(m^*)}}^\pi \{\mathcal{Q} \le \log(M), f(\mathcal{S}_{i*}^\pi) \ne m^*\} &\le \mathbb{P}_{\Psi^{(m^*)}}^\pi \{f(\mathcal{S}_{i*}^\pi) \ne m^*\} \\
&\le \eta_n. \quad (25)
\end{aligned}$$

Combining (23), (24), and (25), we conclude that:

$$\mathbb{P}_{\Psi^{(m^*)}}^\pi \{\mathcal{Q} > \log(M)\} \ge 1 - 2\eta_n. \quad (26)$$

**Analysis of the log-likelihood ratio.** We now show that (26) does not hold when

$$2\frac{T}{n} \cdot D(\boldsymbol{p}, \boldsymbol{\alpha}) < \frac{1}{(1 + \gamma_n)^4} \log\left(\frac{n}{s}\right).$$

(16) is a necessary condition for (15), since every algorithm $\pi$ satisfying (15) has to satisfy (26). Applying Markov inequality,

$$
\begin{aligned}
\mathbb{P}_{\Psi(m^*)}^{\pi}\{\mathcal{Q} > \log(M)\} &\leq \frac{\mathbb{E}_{\Psi(m^*)}^{\pi}[Q]}{\log(M)} \\
&\leq \frac{2(\bar{s}+s+1)T\sum_{k=1}^{K}x_{i^*k}^{\pi}KL(p_{i^*k}, p_{j^*k})}{n\log(M)} \\
&\leq \frac{2(\bar{s}+s+1)T \cdot D(\boldsymbol{p}, \boldsymbol{\alpha})}{n\log(M)} \\
&\leq \frac{1}{(1+\gamma_n)^2},
\end{aligned}
\tag{27}
$$

where the last inequality holds since from the definition of $\gamma_n$,

$$
\log(M) \geq \bar{s}\log\left(\frac{\alpha_{j^*}n}{16e\bar{s}}\right) \geq \frac{1}{1+\gamma_n}\bar{s}\log\left(\frac{n}{s}\right) \quad \text{and} \quad \frac{\bar{s}+s+1}{\bar{s}} \leq 1+\gamma_n.
$$

Thus, we have

$$
\mathbb{P}_{\Psi(i^*)}\{\mathcal{Q} > \log(m)\} \leq 1 - 2\gamma_n(1+o(1)) < 1 - 2\eta_n,
$$

which contradicts (26).

## C   Proof of Lemmas

### C.1   A useful property of the KL divergence

Since $\frac{d^2 KL(a,b)}{da^2} = \frac{1}{a(1-a)}$ and $\frac{1}{\max\{a,b\}} \leq \frac{1}{x(1-x)} \leq \frac{1}{\min\{a(1-a),b(1-b)\}}$ for all $x \in [\min\{a,b\}, \max\{a,b\}]$, we have

$$
\frac{(a-b)^2}{2\max\{a,b\}} \leq KL(a,b) \leq \frac{(a-b)^2}{2\min\{a(1-a), b(1-b)\}}.
$$

Under (A1) and (A2), therefore, there exist positive constants $c_1$ and $c_2$ such that

$$
c_1 p \leq KL(p_{ik}, p_{jk}) \leq c_2 p \quad \text{for all} \quad i, j, k.
$$

### C.2   Proof of Lemma 1

Under (A1) and (A2), we have $KL(p_{ik}, p_{jk}) = O(p)$ for all $i, j, k$. For all $\boldsymbol{\alpha}$ and $\tilde{\boldsymbol{\alpha}}$, we deduce

$$
\begin{aligned}
\Delta(\boldsymbol{x}^*(\boldsymbol{p}, \boldsymbol{\alpha}), \boldsymbol{p}) &\leq \Delta(\boldsymbol{x}', \boldsymbol{p}) + O(p\|\boldsymbol{\alpha} - \tilde{\boldsymbol{\alpha}}\|_2) \\
&\leq \Delta(\boldsymbol{x}^*(\boldsymbol{p}, \tilde{\boldsymbol{\alpha}}), \boldsymbol{p}) + O(p\|\boldsymbol{\alpha} - \tilde{\boldsymbol{\alpha}}\|_2),
\end{aligned}
$$

where $\boldsymbol{x}' = \arg\min_{\boldsymbol{x}\in\mathcal{X}(\tilde{\boldsymbol{\alpha}})}\|\boldsymbol{x} - \boldsymbol{x}^*(\boldsymbol{p}, \boldsymbol{\alpha})\|_2$. Analogously, we have

$$
\Delta(\boldsymbol{x}^*(\boldsymbol{p}, \tilde{\boldsymbol{\alpha}}), \boldsymbol{p}) \leq \Delta(\boldsymbol{x}^*(\boldsymbol{p}, \boldsymbol{\alpha}), \boldsymbol{p}) + O(p\|\boldsymbol{\alpha} - \tilde{\boldsymbol{\alpha}}\|_2).
$$

Therefore, we have

$$
\frac{|\Delta(\boldsymbol{x}^*(\boldsymbol{p}, \boldsymbol{\alpha}), \boldsymbol{p}) - \Delta(\boldsymbol{x}^*(\boldsymbol{p}, \tilde{\boldsymbol{\alpha}}), \boldsymbol{p})|}{\|\tilde{\boldsymbol{\alpha}} - \boldsymbol{\alpha}\|_2} = \frac{|D(\boldsymbol{p}, \boldsymbol{\alpha}) - D(\boldsymbol{p}, \tilde{\boldsymbol{\alpha}})|}{\|\tilde{\boldsymbol{\alpha}} - \boldsymbol{\alpha}\|_2} = O(p).
$$

### C.3   Proof of Lemma 2

Assume that the true clusters $\mathcal{V}_1, \ldots, \mathcal{V}_K$ are given. Since $e(\mathcal{V}_i, \mathcal{V}_j)$ is a sum of $T$ independent Bernoulli random variables, from Chernoff-Hoeffding inequality, for all $1 \leq i \leq K$

$$
\begin{aligned}
&\mathbb{P}\left\{\left|\frac{4e(\mathcal{V}_i, \mathcal{V}_i)}{\delta T}\frac{n(n-1)}{|\mathcal{V}_i|(|\mathcal{V}_i|-1)} - p_{ii}\right| \geq \frac{p}{\sqrt{n}}\right\} \\
&\leq e^{-\frac{\delta T}{4}\cdot KL((p_{ii}+\frac{p}{\sqrt{n}})\frac{|\mathcal{V}_i|(|\mathcal{V}_i|-1)}{n(n-1)}, p_{ii}\frac{|\mathcal{V}_i|(|\mathcal{V}_i|-1)}{n(n-1)})} + e^{-\frac{\delta T}{4}\cdot KL((p_{ii}-\frac{p}{\sqrt{n}})\frac{|\mathcal{V}_i|(|\mathcal{V}_i|-1)}{n(n-1)}, p_{ii}\frac{|\mathcal{V}_i|(|\mathcal{V}_i|-1)}{n(n-1)})} \\
&\leq \exp\left(-\frac{pT/n}{(\log(pT/n))^2}\right)
\end{aligned}
$$

and for all $i, j$ such that $i \neq j$,

$$\mathbb{P}\left(\left|\frac{4e(\mathcal{V}_i, \mathcal{V}_j)}{\delta T}\frac{n(n-1)}{2|\mathcal{V}_i||\mathcal{V}_j|} - p_{ij}\right| \geq \frac{p}{\sqrt{n}}\right)$$

$$\leq e^{-\frac{\delta T}{4} \cdot KL((p_{ij}+\frac{p}{\sqrt{n}})\frac{2|\mathcal{V}_i||\mathcal{V}_j|}{n(n-1)}, p_{ij}\frac{2|\mathcal{V}_i||\mathcal{V}_j|}{n(n-1)})} + e^{-\frac{\delta T}{4} \cdot KL((p_{ij}-\frac{p}{\sqrt{n}})\frac{2|\mathcal{V}_i||\mathcal{V}_j|}{n(n-1)}, p_{ij}\frac{2|\mathcal{V}_i||\mathcal{V}_j|}{n(n-1)})}$$

$$\leq \exp\left(-\frac{pT/n}{(\log(pT/n))^2}\right),$$

since $KL(p, q) \geq \frac{(p-q)^2}{2(p+q)}$.

We now consider $|e(\mathcal{V}_i, \mathcal{V}_j) - e(\mathcal{S}_i, \mathcal{S}_j)|$, the error due to the mis-classified nodes in the first step. Since the observations and the partition from the first step are correlated, we are not able to directly use concentration inequalities for the sum of independent random variables. We thus define a set of partitions so that the first step output $(\mathcal{S}_1, \dots, \mathcal{S}_K)$ belongs to the set where the number of mis-classified nodes is less than $n \exp\left(-C\frac{pT/n}{\log(pT/n)}\right)$ and then show that all partitions in the set satisfy $|e(\mathcal{V}_i, \mathcal{V}_j) - e(\mathcal{S}_i, \mathcal{S}_j)| \leq 2n$ with high probability. We define the sets as follows:

$$\mathcal{A} = \left\{(\mathcal{V}_1', \dots, \mathcal{V}_K') : \cup_{k=1}^K |\mathcal{V}_k \setminus \mathcal{V}_k'| \leq n \exp\left(-C\frac{pT/n}{\log(pT/n)}\right)\right\}.$$

For any given $(\mathcal{V}_1', \dots, \mathcal{V}_K') \in \mathcal{A}$,

$$\mathbb{P}\left(\left|e(\mathcal{V}_i, \mathcal{V}_j)) - e(\mathcal{V}_i', \mathcal{V}_j'))\right| \geq 2n\right)$$

$$\leq \mathbb{P}\left(\left|e(\mathcal{V}_i' \setminus \mathcal{V}_i, \mathcal{V}_j) + e(\mathcal{V}_i \setminus \mathcal{V}_i', \mathcal{V}_j')\right| \geq 2n\right)$$

$$\leq \mathbb{P}\left(e(\mathcal{V}_i' \setminus \mathcal{V}_i, \mathcal{V}_j)) \geq n\right) + \mathbb{P}\left(e(\mathcal{V}_i \setminus \mathcal{V}_i', \mathcal{V}_j')) \geq n\right)$$

$$\leq 2\exp\left(-\frac{n}{6}\right),$$

where the last inequality is obtained from the Chernoff bound since

$$\mathbb{E}\left[e(\mathcal{V}_i' \setminus \mathcal{V}_i, \mathcal{V}_j))\right] \leq p\frac{\delta T}{4}\exp\left(-C\frac{pT/n}{\log(pT/n)}\right) \leq \frac{n}{2}.$$

Since $\binom{a}{b} \leq \left(\frac{ea}{b}\right)^b$,

$$|\mathcal{A}| \leq \left(\begin{matrix} n \\ n\exp\left(-C\frac{pT/n}{\log(pT/n)}\right) \end{matrix}\right) K^{n\exp\left(-C\frac{pT/n}{\log(pT/n)}\right)}$$

$$\leq \left(\frac{en}{n\exp\left(-C\frac{pT/n}{\log(pT/n)}\right)}\right)^{n\exp\left(-C\frac{pT/n}{\log(pT/n)}\right)} K^{n\exp\left(-C\frac{pT/n}{\log(pT/n)}\right)}$$

$$= \exp\left(\left(\frac{CpT}{\log(pT/n)} + n(1 + \log(K))\right)\exp\left(-C\frac{pT/n}{\log(pT/n)}\right)\right)$$

From the union bound, we obtain

$$\max_{i,j}|e(\mathcal{V}_i, \mathcal{V}_j)) - e(\mathcal{S}_i, \mathcal{S}_j))| \leq \max_{(\mathcal{V}_1', \dots, \mathcal{V}_K') \in \mathcal{A}} \max_{i,j}\left|e(\mathcal{V}_i, \mathcal{V}_j)) - e(\mathcal{V}_i', \mathcal{V}_j'))\right| \leq 2n$$

with probability

$$1 - 2|\mathcal{A}|\exp\left(-\frac{n}{6}\right) \geq 1 - \exp\left(-\frac{n}{12}\right).$$

We conclude that with high probability,

$$\frac{|p_{ij} - \hat{p}_{ij}|}{p_{ij}} = O\left(\frac{\log(Tp/n)}{Tp/n} + \frac{1}{\sqrt{n}}\right)$$

## C.4 Proof of Lemma 3

Let $\boldsymbol{x}' = \arg\min_{\boldsymbol{x}\in\mathcal{X}(\hat{\boldsymbol{\alpha}})} \|\boldsymbol{x} - \boldsymbol{x}^*(\boldsymbol{p},\boldsymbol{\alpha})\|_2$ where $\|A\|_2$ is the spectral norm of $A$. When $\left|\cup_{i=1}^K \mathcal{V}_i \setminus \mathcal{S}_i\right| \leq n\exp\left(-C\frac{pT/n}{\log(pT/n)}\right)$, $\frac{|p_{ij}-\hat{p}_{ij}|}{p_{ij}} = O\left(\frac{\log(Tp/n)}{Tp/n} + \frac{1}{\sqrt{n}}\right)$, and $\frac{pT}{n} = \omega(1)$,

$$
\begin{aligned}
\Delta(\boldsymbol{x}^*(\boldsymbol{p},\boldsymbol{\alpha}),\boldsymbol{p}) &\overset{(a)}{\leq} \Delta(\boldsymbol{x}^*(\boldsymbol{p},\boldsymbol{\alpha}),\hat{\boldsymbol{p}}) + O(\max_{i,j}|p_{ij}-\hat{p}_{ij}|)\\
&\overset{(b)}{\leq} \Delta(\boldsymbol{x}',\hat{\boldsymbol{p}}) + O(\max_{i,j}|p_{ij}-\hat{p}_{ij}| + p\max_i|\alpha_i-\hat{\alpha}_i|)\\
&\leq \Delta(\boldsymbol{x}^*(\hat{\boldsymbol{p}},\hat{\boldsymbol{\alpha}}),\hat{\boldsymbol{p}}) + O(\max_{i,j}|p_{ij}-\hat{p}_{ij}| + p\max_i|\alpha_i-\hat{\alpha}_i|)\\
&\overset{(c)}{\leq} \Delta(\boldsymbol{x}^*(\hat{\boldsymbol{p}},\hat{\boldsymbol{\alpha}}),\bar{\boldsymbol{p}}) + O(\max_{i,j}|p_{ij}-\hat{p}_{ij}| + p\max_i|\alpha_i-\hat{\alpha}_i|)\\
&\leq \Delta(\boldsymbol{x}^*(\hat{\boldsymbol{p}},\hat{\boldsymbol{\alpha}}),\bar{\boldsymbol{p}}) + O(p(\frac{\log(Tp/n)}{Tp/n} + \frac{1}{\sqrt{n}})),
\end{aligned}
\tag{28}
$$

where $(a)$ is obtained from $|KL(p_{ik},p_{jk}) - KL(\hat{p}_{ik},\hat{p}_{jk})| = O(\max_{i,j}|p_{ij} - \hat{p}_{ij}|)$ since $\frac{dKL(a,b)}{da} = \log\left(\frac{a(1-b)}{b(1-a)}\right)$ and we assume that $\left|\log\left(\frac{p_{ik}(1-p_{jk})}{p_{jk}(1-p_{ik})}\right)\right| \leq \kappa_U$; $(b)$ is obtained from the fact that $KL(\hat{p}_{ik},\hat{p}_{jk}) = O(p)$ for all $i,j,k$ under (A1); and $(c)$ is obtained from $|KL(\hat{p}_{ik},\hat{p}_{jk}) - KL(\bar{p}_{ik},\bar{p}_{jk})| = O(\max_{i,j}|p_{ij} - \hat{p}_{ij}| + p\max_i|\alpha_i-\hat{\alpha}_i|)$.

Analogously, we can show that

$$
\Delta(\boldsymbol{x}^*(\hat{\boldsymbol{p}},\hat{\boldsymbol{\alpha}}),\bar{\boldsymbol{p}}) \leq \Delta(\boldsymbol{x}^*(\boldsymbol{p},\boldsymbol{\alpha}),\boldsymbol{p}) + O(p(\frac{\log(Tp/n)}{Tp/n} + \frac{1}{\sqrt{n}})).
\tag{29}
$$

Since $\Delta(\boldsymbol{x}^*(\boldsymbol{p},\boldsymbol{\alpha}),\boldsymbol{p}) = \Omega(p)$ under (A2), from (28) and (29), we have

$$
\frac{|\Delta(\boldsymbol{x}^*(\hat{\boldsymbol{p}},\hat{\boldsymbol{\alpha}}),\bar{\boldsymbol{p}}) - \Delta(\boldsymbol{x}^*(\boldsymbol{p},\boldsymbol{\alpha}),\boldsymbol{p})|}{\Delta(\boldsymbol{x}^*(\boldsymbol{p},\boldsymbol{\alpha}),\boldsymbol{p})} = O(\frac{\log(Tp/n)}{Tp/n} + \frac{1}{\sqrt{n}}).
$$

## C.5 Proof of Lemma 4

When $v \in \mathcal{S}_i \cap \mathcal{V}_i$, $e(v,\mathcal{S}_k)$ is a sum of independent Bernoulli r.v. with mean $\bar{p}_{ik}$ with $2(1-\frac{\delta}{2})\hat{x}_{ik}^*\frac{T}{n}$ samples. From the Chernoff-Hoeffding bound, we have

$$
\begin{aligned}
&\mathbb{P}\left(\max_{1\leq k\leq K}\left|e(v,\mathcal{S}_k) - 2(1-\frac{\delta}{2})\hat{x}_{ik}^*\hat{p}_{ik}\frac{T}{n}\right| \geq \frac{\delta}{4}\hat{p}\frac{T}{n}\right)\\
&\leq \sum_{k=1}^K \mathbb{P}\left(\left|e(v,\mathcal{S}_k) - 2(1-\frac{\delta}{2})\hat{x}_{ik}^*\hat{p}_{ik}\frac{T}{n}\right| \geq \frac{\delta}{4}\hat{p}\frac{T}{n}\right)\\
&\leq \sum_{k=1}^K \mathbb{P}\left(\left|e(v,\mathcal{S}_k) - 2(1-\frac{\delta}{2})\hat{x}_{ik}^*\bar{p}_{ik}\frac{T}{n}\right| \geq \frac{\delta}{5}\hat{p}\frac{T}{n}\right)\\
&\leq \sum_{k=1}^K \left(e^{-2(1-\frac{\delta}{2})\hat{x}_{ik}^*\frac{T}{n}KL(\bar{p}_{ik}+\frac{\delta\hat{p}}{10\hat{x}_{ik}^*},\bar{p}_{ik})} + e^{-2(1-\frac{\delta}{2})\hat{x}_{ik}^*\frac{T}{n}KL(\bar{p}_{ik}-\frac{\delta\hat{p}}{10\hat{x}_{ik}^*},\bar{p}_{ik})}\right)\\
&\leq \exp\left(-\frac{pT/n}{(\log(pT/n))^3}\right),
\end{aligned}
$$

where the last inequality is obtained from the facts that $\frac{(a-b)^2}{2\max\{a,b\}} \leq KL(a,b) \leq \frac{(a-b)^2}{2\min\{a(1-a),b(1-b)\}}$ and $\delta = \frac{1}{\log(pT/n)}$. From the above inequality and the Markov inequality,

$$\mathbb{P}\left\{\left|\mathcal{S}_i \setminus \hat{\mathcal{V}}_i\right| \geq \alpha_i n \exp\left(-\frac{pT/n}{(\log(pT/n))^4}\right)\right\}$$

$$\leq \mathbb{P}\left\{\left|\mathcal{S}_i \setminus \mathcal{V}_i\right| + \left|(\mathcal{S}_i \cap \mathcal{V}_i) \setminus \hat{\mathcal{V}}_i\right| \geq \alpha_i n \exp\left(-\frac{pT/n}{(\log(pT/n))^4}\right)\right\}$$

$$\leq \mathbb{P}\left\{\left|(\mathcal{S}_i \cap \mathcal{V}_i) \setminus \hat{\mathcal{V}}_i\right| \geq \frac{\alpha_i n}{2} \exp\left(-\frac{pT/n}{(\log(pT/n))^4}\right)\right\}$$

$$\leq \frac{\mathbb{E}\left[\left|(\mathcal{S}_i \cap \mathcal{V}_i) \setminus \hat{\mathcal{V}}_i\right|\right]}{\frac{\alpha_i n}{2} \exp\left(-\frac{pT/n}{(\log(pT/n))^4}\right)}$$

$$\leq \exp\left(-\frac{pT/n}{2(\log(pT/n))^3}\right) \quad \text{for all} \quad 1 \leq i \leq K.$$

When $v \in \mathcal{S}_i \cap \mathcal{V}_j$, $e(v, \mathcal{S}_k)$ is a sum of independent Bernoulli r.v. with mean $\bar{p}_{jk}$ with $2(1-\frac{\delta}{2})\hat{x}_{ik}^* \frac{T}{n}$ samples. From the Chernoff-Hoeffding bound, we have

$$\mathbb{P}\left(\max_{1 \leq k \leq K}\left|e(v, \mathcal{S}_k) - 2(1-\frac{\delta}{2})\hat{x}_{ik}^*\hat{p}_{ik}\frac{T}{n}\right| \leq \frac{\delta}{4}\hat{p}\frac{T}{n}\right)$$

$$= \prod_{k=1}^{K} \mathbb{P}\left(\left|e(v, \mathcal{S}_k) - 2(1-\frac{\delta}{2})\hat{x}_{ik}^*\hat{p}_{ik}\frac{T}{n}\right| \leq \frac{\delta}{4}\hat{p}\frac{T}{n}\right)$$

$$\leq \prod_{k=1}^{K} \exp\left(-\min_{q \in (\hat{p}_{ik}-\frac{\delta\hat{p}}{4(2-\delta)\hat{x}_{ik}^*}, \hat{p}_{ik}+\frac{\delta\hat{p}}{4(2-\delta)\hat{x}_{ik}^*})}(2-\delta)\hat{x}_{ik}^*\frac{T}{n} \cdot KL(q, \bar{p}_{jk})\right)$$

$$\leq \exp\left(-\sum_{k=1}^{K} 2\hat{x}_{ik}^*\frac{T}{n} \cdot KL(\bar{p}_{ik}, \bar{p}_{jk}) + O\left(\frac{\delta pT}{n}\right)\right)$$

$$\leq \exp\left(-2\frac{T}{n} \cdot D(\boldsymbol{p}, \boldsymbol{\alpha}) + O\left(\frac{\delta pT}{n}\right)\right).$$

From the above inequality and the Markov inequality, with high probability

$$\left|\hat{\mathcal{V}}_i \setminus \mathcal{V}_i\right| \leq \alpha_i n \exp\left(-2\frac{T}{n} \cdot D(\boldsymbol{p}, \boldsymbol{\alpha}) + O\left(\frac{\delta pT}{n}\right)\right) \quad \text{for all} \quad 1 \leq i \leq K.$$

### C.6 Proof of Lemma 5

Since $\frac{|p_{ij}-\hat{p}_{ij}|}{p_{ij}} = O\left(\frac{\log(Tp/n)}{Tp/n} + \frac{1}{\sqrt{n}}\right)$ from Lemma 2, we have $KL(\frac{e(v,\mathcal{S}_k)}{d(v,\mathcal{S}_k)}, \hat{p}_{ik}) \geq \frac{1}{2}KL(\frac{e(v,\mathcal{S}_k)}{d(v,\mathcal{S}_k)}, \bar{p}_{ik})$ under (A1) and (A2). From the Chernoff-Hoeffding bound, we deduce

$$\mathbb{P}\left\{\sum_{k=1}^{K} d(v, \mathcal{S}_k)KL(\frac{e(v,\mathcal{S}_k)}{d(v,\mathcal{S}_k)}, \hat{p}_{ik}) \geq d(v, \mathcal{V})D^R(\hat{\boldsymbol{p}}, \hat{\boldsymbol{\alpha}})\right\}$$

$$\leq \sum_{k=1}^{K} \mathbb{P}\left\{d(v, \mathcal{S}_k)KL(\frac{e(v,\mathcal{S}_k)}{d(v,\mathcal{S}_k)}, \hat{p}_{ik}) \geq \frac{\delta T}{4Kn}\exp\left(\frac{pT/n}{(\log(pT/n))^4}\right)D^R(\hat{\boldsymbol{p}}, \hat{\boldsymbol{\alpha}})\right\}$$

$$\leq \sum_{k=1}^{K} \mathbb{P}\left\{d(v, \mathcal{S}_k)KL(\frac{e(v,\mathcal{S}_k)}{d(v,\mathcal{S}_k)}, \bar{p}_{ik}) \geq \frac{\delta T}{8Kn}\exp\left(\frac{pT/n}{(\log(pT/n))^4}\right)D^R(\hat{\boldsymbol{p}}, \hat{\boldsymbol{\alpha}})\right\}$$

$$\leq 2K\exp\left(-\frac{\delta T}{8Kn}\exp\left(\frac{pT/n}{(\log(pT/n))^4}\right)D^R(\hat{\boldsymbol{p}}, \hat{\boldsymbol{\alpha}})\right).$$

Since we have $D^R(\hat{\boldsymbol{p}}, \hat{\boldsymbol{\alpha}}) = \Omega(p)$ under (A2) and $\delta = \Omega(\frac{1}{\log(pT/n)})$, the above inequality becomes

$$\mathbb{P}\left\{\sum_{k=1}^{K} d(v, \mathcal{S}_k) KL(\frac{e(v, \mathcal{S}_k)}{d(v, \mathcal{S}_k)}, \hat{p}_{ik}) \geq d(v, \mathcal{V}) D^R(\hat{\boldsymbol{p}}, \hat{\boldsymbol{\alpha}})\right\} \leq \exp\left(-\frac{pT \exp\left(\frac{pT/n}{(\log(pT/n))^4}\right)}{n \log(pT/n)^2}\right).$$

Applying Markov inequality, we deduce that this second round additionally generates at most $n \exp\left(-\frac{pT}{n} \exp\left(\frac{pT/n}{(\log(pT/n))^5}\right)\right)$ mis-classified nodes with high probability.

## Footnotes

[1]Formally, the inequality should be $\min_\sigma \mathbb{P}\big[\big|\cup_{k=1}^2 \mathcal{V}_{\sigma(k)} \setminus \mathcal{S}_k^\pi\big| < s\big] \geq 1 - \eta_n$, where the min is over permutations $\sigma$ of $\{1, 2\}$.