[Reviews · NeurIPS 2019]

Reviewer 1



This paper proposes the Adaptive Spectral Partition algorithm for stochastic block models. To the best of the reviewer's knowledge, this adaptive approach is novel for asymptotically exact recovery in the SBMs. Additionally, the authors have showed that compared to regular sampling, this adaptive sampling approach allow one to recover the clusters in a larger regime. The result is interesting to me. Does that imply one is "wasting" some information by using random sampling? The main results seems tight. The authors compare their adaptive sampling algorithm to the regular clustering algorithm on the symmetric SBM with two clusters experimentally. The paper is well written and easy to follow. The proofs employ hypothesis testing, and connect its error rate to the log-likelihood ratio of two constructed models. Overall I believe the paper is a good contribution to the SBM literature.

Reviewer 2



The paper proposes an optimal edge sampling strategy for exact clustering SBM networks which are an important baseline for clustering. The authors find the upper bound on the cluster recovery rate and use those to device a sampling strategy in a spectral algorithm. The authors provide rigorous analysis in their paper for an important problem. It is also a new approach which unlike prior work on SBM clustering, provides a recovery algorithm based on a sampling strategy and not just uses it as a mean to prove claims on clustering feasibility. There are two fundamental things that are problematic in this manuscript – its readability and numerical validation: 1. The readability of the paper: while the authors make some sincere effort to explain some of their main results, the reader is flooded but very many technical statements some of which are not always motivated or explained even intuitively. for example in the presentation fo the Main result in page 2 -x or x_ij are not defined yet. Also, many steps and quantities in the algorithm are not clearly motivate and explained e.g. the setting ot T and delta, p_ii and so on. Hence the reader find himself flooded with quantities which are not straightforward to understand 2. It is stated in the abstract that it is numerically shown that adaptive edge sampling is shown to improve over random. However, no numerical validation or any kind or experimental validation for the sampling strategy is presented in this paper. This is a necessity for such a paper providing an algorithmic solution to a benchmark problem. Post Review: the authors have addressed some of my concerns and intend to modify adequately the final version. I have raised my score accordingly.

Reviewer 3



I have to say I'm not an expert on theorectical adaptive sampling in the SBM. I didn't check the math details, and may not fully understand the math sigficance of the paper. But from my experience on research, I feel this paper is good. The writing is well organized and easy to follow. The studied problem of adaptive sampling is important. The results are comprehensive and promising. The proposed algorithm should be inspiring for practical application.

[Author Response · NeurIPS 2019]

We would like to thank the reviewers for their positive and interesting comments that will help us to enhance our
manuscript. Please find our answers below.

**To Reviewer 1.** *Wasting information under random sampling.* Indeed random sampling is inefficient and waste
information since it probes edges between clusters uniformly; whereas intuitively of course, an optimal adaptive
algorithm gathers 'more' edge information between two clusters that are hard to distinguish.

*Experiments with different a and b.* We thank the reviewer for this nice suggestion. In the revised paper, we will include
a plot showing the proportion of misclassified nodes as a function of a and b, and compare the plot to the theoretical
results. Doing such a figure requires a lot of time, since we have to run our algorithm for a large number of problem
instances. From our past experiences, we are confident that the plot will match and illustrate the theoretical results well.

*Complexity of the ASP algorithm.* Thanks for this question that we should have answered in the paper. The complexity
of the ASP algorithm is polynomial to both $n$ and $T$. Indeed, Step 1 (see Algorithm 1), including the Spectral Clustering
Algorithm, requires $O(T \log(n))$ operations. Step 2 requires $O(T)$ operations to estimate parameters and Step 3
solves a linear program where the number of variables is $k^2$ which does not scale with $n$ and $T$. The remaining steps
simply check the log-likelihood values of each node, which requires $O(T)$ computations. Overall, the computational
complexity of ASP is $O(T \log n)$.

*Derivation of the bound on Page 2 from the main theorem.* Let $p = p_{11} = p_{22}$ and $q = p_{12} = p_{21}$. When
$KL(p,q) \geq KL(q,p)$, $D(\boldsymbol{p}, \boldsymbol{\alpha}) = 2KL(p,q)$. Since both $p = \frac{a \log(n)}{n}$ and $q = \frac{b \log(n)}{n}$ are $o(1)$, we can derive
$\frac{KL(p,q)}{\log(n)/n} = a \log(\frac{a}{b}) + (b-a)(1+o(1))$. Therefore, $\frac{nD(\boldsymbol{p},\boldsymbol{\alpha})}{\log(n)} \geq 1$ if and only if $a \log(\frac{a}{b}) + (b-a) \geq 1$. Analogously,
we can conclude $b \log(\frac{b}{a}) + (a-b) \geq 1$ is equivalent to $\frac{nD(\boldsymbol{p},\boldsymbol{\alpha})}{\log(n)} \geq 1$ when $KL(p,q) \leq KL(q,p)$. We will add this
discussion to the revised paper.

*Which clustering algorithm did the author use for the red group in Figure 1?* The spectral partition algorithm described
in [16] is used. This algorithm is proved to be optimal in terms of error rate in [16].

**To Reviewer 2.** *Readability of the paper.* The paper is on the theoretical side, and requires rather technical and novel
proofs. We agree that for readers not familiar to the stochastic block model and spectral clustering methods, it may be
difficult to follow. We will make significant efforts to simplify when appropriate. Note that Rev. 1 and 3 found the
paper easy to read and well written, but again, it depends on the reader's background.

*Missing definitions and motivations.* Note that in Page 2, $x_{ij}$ is just a "dummy" variable for the optimization problem
$D(\boldsymbol{p}, \boldsymbol{\alpha})$, so it does not require any definition. Later in Page 3, we provide an insightful interpretation of $x_{ij}$. In each
subsection of Section 4.2, we provide the motivation of the corresponding step of Algorithm 1. For instance, Section
4.2.1 explains how to divide the total sampling budget $T$ and Section 4.2.2 states the meaning of $\hat{\boldsymbol{p}}$. We will extend
these parts to motivate the algorithm in even more detail.

*Numerical validation.* The right graph of Figure 1 in our main manuscript Page 3, we compare the error rates of the
adaptive spectral partition to that of a non-adaptive spectral partition algorithm described in [16] known to be optimal
(in absence of adaptive sampling). We agree that the paper would benefit from more numerical results and will add
more experiments in the revised paper.

**To Reviewer 3.** To explain the novelty of our contributions, we can say that:
1. the paper derives for the first time necessary and sufficient conditions for both asymptotically accurate detection and
exact recovery in the Stochastic Block Model (SBM) with adaptive sampling. This was an important open problem in
the community interested in the SBM.
2. Our proof techniques are novel and also allow for the first time the derivation of a necessary and sufficient condition
holding *with high probability* (see the discussion the paragraph "Deriving fundamental limits" on Page 3 for an
explanation why it is challenging). Note that most often, researchers are able to derive fundamental limits for the
expected number of misclassified nodes, and devise an algorithm with performance guarantees holding with high
probability (with probability tending to 1 as n goes large, the number of misclassified nodes is small).
In summary, in this paper, we manage not only to deal with adaptive sampling, but we also fix the aforementioned gap
between fundamental limits and performance guarantees (both hold with high probability). We will emphasize and state
the contributions more clearly in the revised paper.

[Meta-Review · NeurIPS 2019]

This paper proposes a new adaptive sampling and clustering framework for learning stochastic block models for network data. The paper provides a comprehensive theoretical analysis of the framework, as well as empirical results, that clearly demonstrate the useful contributions of the approach. Reviewers were in agreement that the paper is worthy of acceptance for the conference. There were some comments from reviewers about opportunities to improve the presentation in the paper. The authors are strongly encouraged to take these reviewer comments into account when revising the paper for the final version.